# Fast DL-based Simulation with Microarchitecture Agnostic Traces and Instruction Embeddings

Santosh Pandey*, Amir Yazdanbakhsh†, Hang Liu*

\* Rutgers University. *santosh.pandey@rutgers.edu,hl1097@soe.rutgers.edu*

† Google DeepMind. *ayazdan@google.com*

*Abstract*—**Microarchitecture simulators are indispensable tools for microarchitecture designers to validate, estimate, optimize, and manufacture new hardware that meets specific design requirements. While the quest for a fast, accurate and detailed microarchitecture simulation has been ongoing for decades, existing simulators excel and fall short at different aspects: (i) Although execution-driven simulation is accurate and detailed, it is extremely slow and requires expert-level experience to design. (ii) Trace-driven simulation reuses the execution traces in pursuit of fast simulation but faces accuracy concerns and fails to achieve significant speedup. (iii) Emerging deep learning (DL)-based simulations are remarkably fast and have acceptable accuracy, but fail to provide adequate low-level microarchitectural performance metrics such as branch mispredictions or cache misses, which is crucial for microarchitectural bottleneck analysis. Additionally, they introduce substantial overheads from trace regeneration and model re-training when simulating a new microarchitecture.**

**Re-thinking the advantages and limitations of the aforementioned three mainstream simulation paradigms, this paper introduces TAO that redesigns the DL-based simulation with three primary contributions: First, we propose a new training dataset design such that the subsequent simulation (i.e., inference) only needs functional trace as inputs, which can be rapidly generated and reused across microarchitectures. Second, to increase the detail of the simulation, we redesign the input features and the DL model using self-attention to support predicting various performance metrics of interest. Third, we propose techniques to train a microarchitecture agnostic embedding layer that enables fast transfer learning between different microarchitectural configurations and effectively reduces the re-training overhead of conventional DL-based simulators. TAO can predict various performance metrics of interest, significantly reduce the simulation time, and maintain similar simulation accuracy as state-of-the-art DL-based endeavors. Our extensive evaluation shows TAO can reduce the overall training and simulation time by 18.06× over the state-of-the-art DL-based endeavors.**

## I. INTRODUCTION

Since its inception, microarchitecture simulators rapidly become the most commonly used tools in computer architecture-related research (see the report [37]). As of today, computer architecture simulation is the textbook standard and virtually used in any architecture explorations, e.g., design space exploration [18], [22], [23], [40], microarchitectural bottleneck analysis [5], [14], workload characterization [16], [30] among many others [25], [41]. As a common practice, architecture researchers often use popular software architecture simulators to incorporate their radical new ideas. The simulation yields a range of metrics that characterize the execution of benchmarks, with the level of detail in the simulation dictating the specificity of these metrics. Such output metrics provide feedback to the researcher for further explorations and/or decision makings.

The quest towards a *fast*, *accurate* and *detailed* cycle-level architecture simulation has never stopped. This cohort of researchers have mainly dedicated their efforts into three prominent paradigms, i.e., execution-driven simulation [1], [4], [7], [19], [29], [31], [34], [41], trace-driven simulation [2], [3], [13], [20], [21], [33], and recently the DL-based simulation [24], [28], [32], [38] (see [6], [8], [11], [39] for more types of architecture simulations). Figure 1(a)-(c) illustrates the workflow used by each paradigm: (i) Execution-driven simulation offers the most detailed and accurate framework, although this comes at the cost of extremely slow speed and high maintenance overhead. (ii) Trace-driven simulation faces accuracy concerns in pursuit of higher throughput than execution-driven simulations with trace reuse. Reusing the same trace for different microarchitectures raises accuracy concerns as the execution order of different memory instructions can vary. (iii) Emerging DL-based simulations are remarkably fast and can provide comparable cycle-level accuracy, whereas hit three roadblocks: limited output metrics, expensive microarchitecture-specific trace generation, and restricted microarchitecture support.

Departing from the designs and desired goals from the aforementioned three paradigms, this paper redesigns DL-based cycle-level microarchitecture simulator. Particularly, we take as input the functional and detailed traces train a DL-based simulator, support a set of desired performance metrics of interest and fast microarchitecture exploration, achieving the comparable accuracy as execution-driven simulation, and an order of magnitude higher throughput than the state-of-the-art DL-driven simulator, i.e., SimNet. Figure 1(d) illustrates the workflow of our system, which encompasses the following three contributions:

- First, we introduce a unique training dataset design so that the subsequent simulation (i.e., inference) only needs light-weighted and reusable functional trace as inputs.

- Second, for predicting a variety of performance metrics of interest, we propose an DL model with separate embedding and self-attention based performance prediction layers.

- Third, we introduce transfer learning techniques to rapidly explore various microarchitectures. This includes

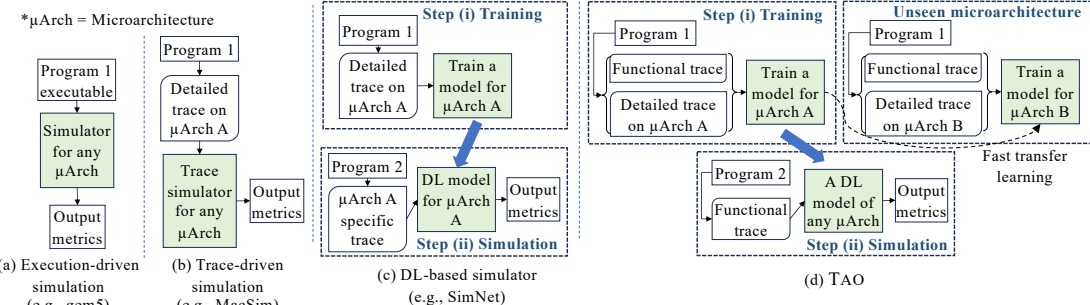

Fig. 1: Mainstream simulation mechanisms vs. our effort, i.e., TAO.

architecture agnostic embedding layers and judicious training dataset selection.

## II. DESIGN PRINCIPLE, CHALLENGE AND OVERVIEW

This paper adheres to two design principles for designing DL-based simulator. *Design Principle #1*. We advocate that (i) the input to the DL model should only capture the instruction execution sequence and (ii) the DL model should govern the hardware features. *Design Principle #2*. An DL-based microarchitecture simulator should (i) report various performance metrics during the architecture simulation and (ii) support rapid explorations of different architecture configurations.

**Challenges.** TAO faces three grand challenges: (i) For the training dataset, we need to associate the microarchitecture impacts with each executed instruction in the functional trace. (ii) Reporting various performance metrics demands us to derive sufficiently powerful DL models that can capture the impacts of various hardware components. (iii) Training microarchitecture-agnostic program embeddings presents difficulties because the embeddings are biased towards the architecture they are trained on. These three challenges motivate the design of TAO.

**Overview.** Section IV unveils TAO, our multi-modal DL architecture for microarchitecture simulation. Our approach adheres to design principle #1 by proposing a workflow to construct training datasets from detailed and functional traces which attributes the differences in these two traces to performance metrics, allowing the reuse of functional traces for varying microarchitectures. For design principle #2, we propose multi-metric predictions with feature engineering with a self-attention model to increase the simulation detail. Further, we propose techniques to train microarchitecture agnostic embedding layers that enable fast transfer learning which significantly reduces the re-training overhead of DL-based microarchitecture training and simulation.

## III. BACKGROUND

**Execution trace.** This paper extensively uses execution trace, which refers to the stream of instructions generated by functional or detailed simulation. The gem5 simulator is modified to generate execution traces capturing various static instruction properties and dynamic performance metrics. Functional trace refers to the microarchitecture agnostic trace generated with functional simulation using *AtomicSimpleCPU*

model. We use the terms functional trace and microarchitecture agnostic trace interchangeably. It only contains static properties like opcode, registers, and other instruction flags. Detailed trace refers to the trace generated with the *O3CPU* model. It captures various microarchitecture specific performance metrics like data access misses, instruction cache misses, branch mispredictions, speculative instructions and latency of individual instructions.

## IV. TAO

### A. Training Dataset Construction

TAO uses functional trace as input to the model and the output (i.e. label) can be various performance metrics. This permits the subsequent simulation (i.e., inference) to only require functional trace as inputs, which can be rapidly generated and reused across microarchitectures. For the output, we use three major performance metrics, i.e., latency, branch misprediction, and data cache misses, to explain how we process the detail and function traces to arrive at the training dataset. However, it is important to note that TAO can potentially support other performance metrics.

Functional and detailed traces output similar sequence order, which permits us to associate each instruction of a functional trace with a detailed trace. However, the challenge is that the difference in number of instructions between detailed and functional traces is nontrivial. Detailed trace generally differs from functional trace in the following two aspects. The detailed trace includes two types of additional dynamic instructions during execution that are missing in the functional trace. Specifically, the detailed trace contains incorrect speculative and stall instructions. Incorrect speculative instructions are the wrongly executed instructions squashed based on branch prediction. Stall instructions are used to stall the pipeline by inserting a no-operation (nop) instruction in the pipeline when any other instructions cannot be executed. Our key idea is that both types of additional instructions can be converted into numerical performance differences and attributed to specific instructions from the functional trace.

**Squashed speculative instructions.** If the predicted branch path is correct, speculatively executed instructions will be correct, thus the instruction streams of detailed and functional traces will be identical. When a speculative path is wrong due to branch misprediction, speculatively executed instructions should be squashed. This case leads to a distinction between

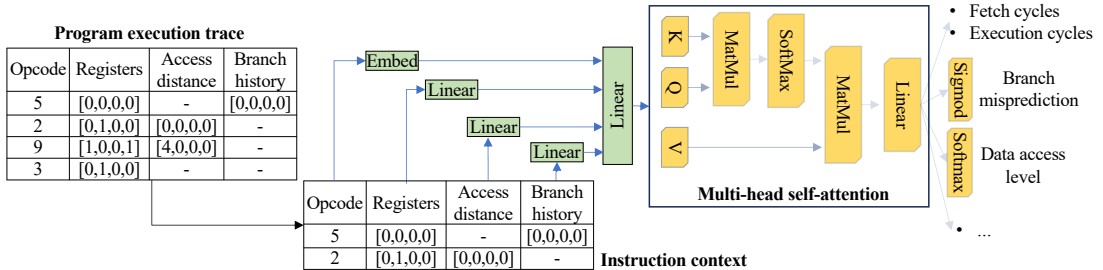

Fig. 2: Our initial DL model architecture.

functional and detailed traces. The total impact of branch misprediction can be accounted for in the functional trace with the fetch timing information obtained from the detailed trace. If a branch is mispredicted, it will delay the fetch of the next correct instruction. In a detailed trace, the fetch latency of the correct instruction does not include the speculation or branch resolution overhead. To include the miss prediction overhead, we remove the squashed instruction from the detailed trace, get the difference in the fetch clock as the fetch latency, and add it to the subsequent instruction.

**Pipeline stalls.** Stall instructions can be handled similarly to squashed speculative instructions. When no instruction can be executed in the pipeline due to dependency or resource contention, `nop` instructions are filled. Similar to squashed speculative instructions, we remove and project the latency impact of `nop` instructions to the subsequent instruction. We use the fetch clock from the detailed trace to determine the additional fetch latency delay.

### B. Multi-Metric DL Model Design

**Feature engineering.** We propose new techniques to build cross-instruction features, in addition to the per-instruction features from the state-of-the-art [24]. We extract four key instruction properties from the microarchitecture agnostic execution trace: the opcode, registers, data access address and PC address. Opcode and registers derive the per-instruction features. For opcode, we employ an integer mapping for each unique opcode in the dataset. Regarding registers, since the instructions can involve multiple registers, we create a bitmap vector with a size equal to the total number of registers. If an instruction uses $i_{th}$ register, $i_{th}$ index in the vector will be set to 1 (0 otherwise). Both source and destination registers are included in the bitmap vector.

Cross-instruction features are derived from the PC and memory addresses. We use the branch history as input to model the outcome of conditional branch instructions. This history, indicating the outcomes of prior branch instructions, is employed by existing branch predictors to predict whether the branch will be taken [15]. To model the data access level, we calculate the access distance, which is the difference between current memory access and the previous $N_m$ memory accesses. We use a memory context queue to track the access distance of $N_m$ memory accesses.

**DL model architecture.** Figure 2 exemplifies our DL model design. The model first generates instruction embeddings from

input features with two-level embedding layers and then uses multi-headed self-attention to perform multi-metric prediction. We use a sequence of $N+1$ instructions as input to the model. The embedding layers generate instruction embeddings in two steps. Initially, embeddings are created independently for each category of input. This separate generation facilitates enhanced representation learning for each category. Specifically, for opcode, a trainable lookup table based embedding layer is employed. For the remaining categories, distinct linear embedding layers are utilized. The individual instruction embedding is obtained by combining categorical embeddings through a linear layer. Note embedding layers independently generate instruction embeddings for current and $N$ context instructions.

Following the generation of instruction embeddings, the prediction layers employ multi-head self-attention to determine the performance metrics. Considering the impact of microarchitecture, this approach allows attention layers to model the interaction between current and earlier instructions. Using self-attention obviates the need for manually tracking context instructions, enhancing efficiency. Employing multiple heads enables each head to learn unique hardware-instruction interplay. The output from each head is concatenated and passed through a linear layer. Overall, the model predicts the latency for each instruction individually.

We use different operators to predict different performance metrics based on the output of the last linear layer: (i) The fetch and execution cycles are directly predicted from the linear layer. (ii) An additional sigmoid layer is incorporated for branch prediction to predict whether the branch will be mispredicted. (iii) We use a softmax layer for the data access level, as the output can be multiple categories. (iv) More performance metrics like instruction cache miss and TLB miss can be predicted through a sigmoid layer. During training, a loss is computed from each performance metric and combined with a linear ratio in backpropagation. To obtain the total cycle of all instructions, we use the retire clock of instructions. Retire clock is computed as current clock + fetch latency + execution latency. The retire clock of the last instruction of a benchmark determines the total cycles.

**Intuitive explanation on supporting a set of performance metrics.** Multi-metric prediction exploits the relatedness of performance metrics. With the attention model and microarchitecture agnostic input, our design allows us to output various performance metrics of interest. It can capture the relation-

ship between each performance metric and the specific input features that impact the metric. This allows all metrics to be derived from the same hidden layers. Multi-metric prediction has two benefits. First, it increases the output details of the simulation. Second, individual loss from data access level and branch prediction helps the model relate the cycle prediction with memory and branch behavior during training.

### C. Fast Transfer Learning via Microarchitecture Agnostic Embeddings

Figure 3 illustrates our fast transfer learning process to enable TAO for a new unseen microarchitecture rapidly, i.e., $\mu$Arch C, employing microarchitecture agnostic embedding layers and fine-tuning. Initially, shared embedding layers are trained with two carefully selected microarchitectures, i.e., $\mu$Arch A and $\mu$Arch B. During training for $\mu$Arch C, the parameters of shared embedding layers are frozen, i.e., we do not update the parameters during backpropagation. The parameters of prediction layers and embedding adaptation layer are fine-tuned with the training dataset for $\mu$Arch C.

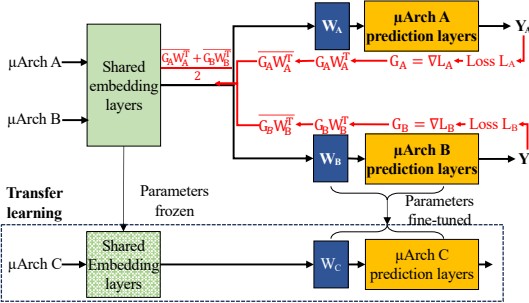

Fig. 3: Overview of transfer learning process for microarchitecture $\mu$Arch C and gradient normalization of TAO.

**Microarchitecture agnostic embedding design.** The shared embedding layers generate embedding for each individual instruction, and microarchitecture specific prediction layers predict the performance labels. The prediction layers of each microarchitecture computes the gradients for the embedding layers separately. We propose to combine them to update the shared embedding layers.

Such designs that combine gradients to update shared layers can face two critical issues: *negative transfer* and *imbalance in gradient magnitude* for shared layers: (i) Negative transfer [26], [43] occurs when the shared layers receive gradients from different microarchitecture that are opposite to each other. (ii) Imbalance in gradients magnitude [12] arises when one microarchitecture is too dominant during training, inducing gradients with relatively large magnitudes. These issues impact convergence and generalization [12], [43].

In Granite, to derive the gradients for shared embedding layers, the gradients from the prediction layers of each $\mu Arch$ are averaged. Just averaging the gradients may resolve neither the negative transfer nor gradient imbalance problem [42]. Using gradient imbalance as an example, if the gradient of one task is larger in magnitude than the other, the larger one will dominate the average gradients. GradNorm, addresses the imbalance in gradient magnitude for multi-task learning by using learnable combination weights ($w_A$ and $w_B$) to combine the losses from each task. This indirectly controls the magnitude of the gradients. The underlying rationale is to dynamically adjust the combination weights in response to the gradient magnitudes of shared layers, ensuring they neither become excessively large nor too small.

While GradNorm can address gradient magnitude imbalance, it cannot adequately address negative transfer issues that arise from conflicting gradient directions. Of note, conflicting gradients may appear when the performances of two different microarchitectures are opposite for the same instruction. Modifying the magnitude of gradients may not effectively change gradient direction in joint training [42]. Hence, it may not fully mitigate the adversarial effect of gradients. Adding this linear projection layer resolves the negative transfer issue as follows: during backpropagation, to compute the gradients for the linear projection layer, we multiply the gradients from the earlier layer $G_A$ with the transpose of the weight matrix $W_A$, i.e., $G_A W_A^T$ based on the chain rule. Under most of the cases, this operation rotates the gradients in the gradient space, changing the direction of gradients.

Figure 3 illustrates our design that tackles negative transfer and gradient imbalance. In contrast to GradNorm which relies on reactive approaches of projecting conflicting gradients to a different plane [42] or finding common direction [35] to mitigate negative transfer, we adopt a proactive solution. We add an individual embedding adaptation layer, i.e., $W_A$ for $\mu$Arch A, similarly $W_B$ for $\mu$Arch B, between the embedding and performance network. The linear layer $W_A$ projects the shared embedding (i.e., Green layers) into microarchitecture specific spaces (i.e., $\mu$Arch prediction layers) during forward propagation. To tackle the gradient imbalance concern, we normalize the gradients for the embedding layers based on the magnitude of the gradients $\overline{G_A W_A}$ and $\overline{G_B W_B}$ to reduce any existing gradient magnitude imbalance.

**Training dataset**. TAO only uses two microarchitectures based on performance variations to train the model efficiently with the desired accuracy. This is significantly more efficient than training general embedding layers with random microarchitectures. To achieve the accuracy and efficiency goal, we define metrics to measure the architectural variations and select the two architectural variations with the most difference.

To measure the microarchitecture variations, we select four performance metrics, i.e., CPI, L1 cache miss, L2 cache miss, and branch misprediction rate. We measure the performance metrics difference of different microarchitectures with Mahalanobis distance [27] instead of Euclidean or Cosine distance for two reasons: (i) Euclidean distance is sensitive to a larger value of one metric, and Cosine distance ignores the value difference. (ii) The other two distances do not consider the correlation among the performance metrics or their scales during distance computation.

| | | **TAO** | **SimNet** | Speedup (vs. SimNet) | **gem5** | Speedup (vs. gem5) |
|---|---|---|---|---|---|---|
| Training | | 1.9 hours | 54.2 hours | 28.52× | - | - |
| Simulation | Trace generation | 0.53 hours | 13.22 hours | 24.94× | 7.81× 14.01 hours | 7.26× |
| | Inference | 1.41 hours | 1.93 hours | 1.37× | | |
| Overall | | 3.84 hours | 69.35 hours | 18.06× | 14.01 hours | 3.66× |

TABLE I: Simulation time comparison with the state-of-the-art DL-based simulator SimNet and gem5 for 10 billion instructions.

## V. EVALUATION

We use the widely adopted SPEC CPU2017 [10] benchmark suite to evaluate TAO. To construct the training dataset, we first generate program traces of 100 million instructions from each training benchmark with default test workloads using the gem5 O3CPU and AtomicSimpleCPU model, respectively. We study the simulation error and throughput in this section. Particularly, simulation error represents the absolute CPI prediction error for each benchmark and is defined as $\frac{|CPI_{pred} - CPI_{truth}|}{CPI_{truth}} \times 100\%$. Simulation throughput is measured in million instructions per second (MIPS).

### A. Comparison with the State-of-the-Art

Our evaluation of accuracy shows that for most microarchitectures and benchmarks, TAO closely matches the simulation error of SimNet. On average, SimNet and TAO exhibit simulation errors of 5.11% and 5.23%, respectively. The slightly higher simulation error of TAo can be attributed to prediction error for branch misprediction and cache misses.

Table I compares the overall time for training and simulation of SimNet vs TAO. Both SimNet and TAO are trained until the error during training is under 6%. It takes 54.2 hours to train a CNN SimNet model. Meanwhile, with microarchitecture agnostic embeddings and transfer learning, TAO can train a model with similar accuracy in merely 1.9 hours. It improves the training time by 28.52×. For simulation, SimNet requires 13.22 hours to generate an input trace with 10 billion instructions. In contrast, utilizing the microarchitecture independent trace, the trace generation time is significantly reduced to 0.53 hours for TAO. For SimNet, it takes 1.93 hours to simulate 10 billion instructions with a simulation throughput of 1.46 MIPS. On the other hand, TAO completes the simulation in 1.41 hours with a throughput of 1.98 MIPS. This speedup against SimNet results from the use of inexpensive functional trace and avoiding history context simulation involving CPU-GPU data movements. Overall, TAO demonstrates a remarkable speed advantage for simulating a new microarchitecture, being 18.06× faster than SimNet. Compared to gem5, TAO provides a speedup of 3.66×.

### B. Multi-metric Prediction Error

This section compares the prediction error for L1 cache miss and branch misprediction. For evaluation, we vary the L1 Dcache size (16KB, 32KB, 64KB, 128KB) and the branch predictors (`Local`, `Tournament`, `BiMode`, and `TAGE_SC_L`).

Figures 4(a) compares the average cache MPKI across four test benchmarks obtained while varying L1 DCache size for gem5 simulation and TAO. The simulated cache MPKI

decreases as the cache size increases from 16KB to 128KB. Cache MPKI predicted by TAO aligns with the simulated result from gem5, demonstrating that a cache size of 128KB results in the least MPKI.

In Figures 4(b), we compare the average branch MPKI across four test benchmarks using different branch predictors for gem5 and TAO. The simulated result from gem5 indicates the highest branch MPKI for the `Local` and the lowest for the `Tage_SC_L`. Branch MPKI predicted by TAO also aligns with the simulated result from gem5. The prediction error is lower for simpler branch predictors like `Local`, experiencing only a marginal increase for relatively complex branch predictors like `Tage_SC_L`. Nonetheless, TAO maintains the relative accuracy across the spectrum of branch predictors.

## VI. GENERALITY OF TAO

**Unseen benchmarks.** TAO can be generalized across a wide variety of unseen benchmarks. The generality of TAO comes from the fact that the deep learning model is trained at the instruction level. We use multiple diverse training benchmarks to train over a variety of instructions. That allows TAO to predict performance metrics for each instruction across different benchmarks accurately. Our evaluation confirms that TAO maintains a good accuracy over unseen benchmarks.

**Unseen architectures.** TAO is designed to simulate single-core out-of-order superscalar processors. To simulate an unseen microarchitecture, we gather a training dataset through gem5 simulation and train the DL model with transfer learning (see Figure 1(d)). TAO can accommodate changes in ISAs similarly to microarchitecture changes with some additional feature engineering for ISA-specific opcodes and registers. TAO cannot be directly used to simulate multi-core CPU and GPU architectures. However, the techniques proposed in this paper, i.e., microarchitecture agnostic trace, embeddings, and multi-metric prediction, establish a framework for a rapid DL-based simulation and is transferable to other architectures.

## VII. RELATED WORKS

Earlier ML-based performance models [17], [36] opt to build a performance model that can extrapolate the performance to unseen microarchitecture designs by simulating a few designs. Recent DL-based prediction models overcome the limitations of ML-based performance models by increasing the level of abstraction at the instruction level. Ithemal [28] and Granite [38] are two recent works performing basic block prediction. These models first gather training datasets by collecting the basic blocks with tools like Dynamorio [9]. The models predict the latency of each block separately. The input features are constructed based on each instruction and

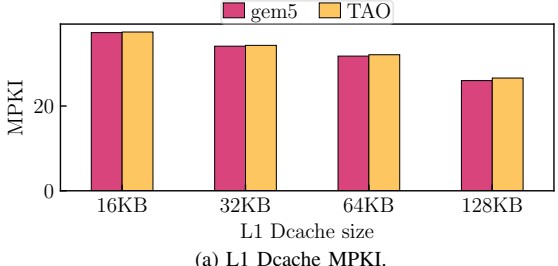 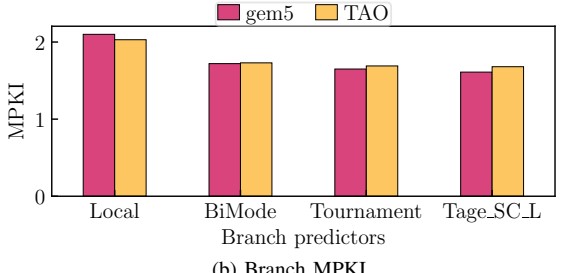

(a) L1 Dcache MPKI.  (b) Branch MPKI.

Fig. 4: Prediction error of TAO for L1 Dcache misses and branch mispredictions.

their structure in the basic blocks. Ithemal uses LSTM to construct the embeddings for each basic block hierarchically. Meanwhile, Granite leverages the structure and dependency graph of instructions within the basic block and GNN models for throughput prediction. Basic block throughput prediction models are limited to static basic block prediction, ignoring the impact of caches and branch prediction.

## VIII. CONCLUSION AND FUTURE WORK

This paper introduces a DL-based simulator TAO that supports detailed, accurate and fast microarchitecture explorations. Notably, it achieves $18.06\times$ higher throughput compared to the state-of-the-art simulator, i.e., SimNet.

While TAO notably enhances simulation throughput, further research is required to enhance the practicality of DL-based simulation. DL-based simulators currently rely on traditional detailed simulations to gather training datasets for unseen microarchitectures, incurring overhead in simulator design. Futhermore, current DL-based simulation is confined to single-core architectures, highlighting the need for research into modeling inter-core interaction and synchronization for multi-core architectures. Addressing these shortcomings could make DL-based simulation more appealing for architects.

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
