# OpenReview forum: "Fast DL-based Simulation with Microarchitecture Agnostic Traces and Instruction Embeddings"
_iscaconf.org/ISCA/2024/Workshop/MLArchSys — MLArchSys 2024 OralPoster_

### Official Review · Reviewer_ENUv · 2024-05-24
**Paper about using multi-headed attention to improve DL based uarchitecture simulations**

**Confidence:** 3
**Rating:** 6

**Detailed Feedback And Questions For Authors:**

This paper presents work on improving Deep Learning based micro-architecture simulators via multi-headed self-attention to model a correlation between the uarch features and to design a uarch agnostic simulator in general. Paper provides clear motivations and explanations, except couple of questions on scalability and practicality that were addressed at  "top reasons to reject the paper". In general, I think that TAO is an interesting paper that can spur some discussions about further usage of transformer models for uarch simulation between the participants of the workshops and wider community, hence I support acceptance of the paper.

**Top Reasons To Accept The Paper:**

1) Huge speedup numbers over baseline implementations.
2) Motivations and implementation explanations are clear.
3) Interesting insights about using attention to improve correlation between uarchitectural features and performance numbers that can lead to some useful discussions within a community.
4) Capability for a uArchitecture agnostic transfer learning.

**Top Reasons To Reject The Paper:**

1) Solution can not expand straightforwardly for analyzing multi-core architectures, NUMA memory and more complicated coherency and consistency models which are practical state of the art hardware.
2) Not clear how cache replacement or related dynamic features can be modeled by their formula for memory access distance.
3) Despite the fact that, they have comments on controlling magnitude of values of different features and performance numbers for different uArch, it is not clear what and how they particularly fine tune features for transfer learning. Is it about only negative and absolute value manipulation that they mention, or is there any other clever stuff?

---

### Official Review · Reviewer_wX4a · 2024-05-26
**Good approach, lacks clarity**

**Confidence:** 3
**Rating:** 5

**Detailed Feedback And Questions For Authors:**

Overall, the approach is novel, however, the lack of clarity in many places makes it very hard to follow the paper. I have the following set of questions for the authors:

1. When you compare detailed and functional traces, you say that detailed traces include various performance metrics. What kind of trace is it? Is it not an instruction trace? Can you provide an example of this trace?
2. In figure 2, why is instruction context a subset of execution trace? Probably  it is related to the  statement "we use a sequence of N+1 instructions as input to the model". However, nowhere this is clarified and nowhere is N defined. Kindly clarify
3. Are N_m memory accesses unique?
4. How is the negative transfer solved using the embedding adaptation layer? The explanation on page 4 right column top paragraph only pertains to solving the gradient imbalance problem.
5.  What does the simulation error in section IV correspond to? Is it the combined error (what kind of mean?) in all the predicted metrics or something else?
6. What is meant by history context simulation involving CPU-GPU data movements? Is it used in SimNet?

**Top Reasons To Accept The Paper:**

1. Relevant work towards fast, cycle-level simulation
2. Provides relevant output metrics with light-weight microarchitecture agnostic functional traces

**Top Reasons To Reject The Paper:**

1. The evaluation is shallow
2. The paper lacks clarity in many places

---

### Official Review · Reviewer_tSsq · 2024-05-27
**Needs more technical details and clarification**

**Confidence:** 3
**Rating:** 4

**Detailed Feedback And Questions For Authors:**

This paper presents an ML-based simulation methodology targeting CPUs. Although the approach is interesting, many technical details need more clarifications.

- Some definitions could be clarified. For example, more precise definitions of "functional trace" and "detailed trace" could be provided.

- The bitmap-based register representation doesn't seem to be distinguishing between source and destination registers. Why this works could be further discussed.

- The ISA used in the evaluation could be clarified. Also, the coverage of the proposed model (RISC only or flexible to cover CISC as well) could be clarified.

- The "squashed speculation instructions" pipeline stall modeling methodology needs more clarification. Because the information is provided only from a high level (basically, utilizing detailed traces), many details are unclear. For example, it is unclear how TAO handles variants of forwarding logics for data hazard resolution.

**Top Reasons To Accept The Paper:**

- Fast training and simulation time

**Top Reasons To Reject The Paper:**

- Many technical details need to be clarified
- The methodology requires to analyze detailed traces (e.g., "squashing" for speculative and stalls), which requires to run time-consuming detailed simulations.
- The scope is limited to single-core CPUs.

---

### Official Review · Reviewer_sZdr · 2024-05-29
**TAO: Re-Thinking DL-based Microarchitecture Simulation**

**Confidence:** 4
**Rating:** 7

**Detailed Feedback And Questions For Authors:**

# Summary
* This paper presents TAO a neural network based microarchitecture simulator.
* The authors present the network architecture of TAO, which comprise of a single layer of multi-head
attention which is capable of predicting multiple microarchitecture metric.
* The authors claim that the weights trained on a specific microarchitecture is easily transferrable
to other microarchitectures via fine tuning.

# Comments
Thank you for submitting your work to MlArchSys. I liked the paper and found the idea of using a single
network to predict multiple microarchitecture features very interesting. I have some clarification and
suggested improvements listed below. I hope you find this helpful for further improving your submission.

1. The network proposed in TAO is depicted as the cost model it is unclear from the text of the paper
on how a state of the system would be simulated solely using TAO.
2. It is unclear what the input and the output space of the proposed model is. From the text the reader
can estimate the the entire sequence of instructions is fed to the model as a vector, however it is not
clear whether the model estimates the performance metrics for each instruction or once for the entire
sequence of instructions.
3. A suggested use case highlighting the deployment of the tool would also be useful for the reader to
appreciate the contribution. For example it is not clear if one needs to write a new simulator to train
TAO if the users are exploring a new architecture idea, or would TAO help them perform the exploration by itself.

**Top Reasons To Accept The Paper:**

1. Interesting approach of using a single network to model different cost metrics.
2. The transfer learning capability as highlighted by the authors is useful since the model could be
adopted for newer microarchitecture without extensive training dataset creation.
3. The prediction and training speedup over the current state of the art is significant.

**Top Reasons To Reject The Paper:**

NA

---

### Official Review · Reviewer_3jMi · 2024-05-29
**An improved approach to leveraging DL for microarchitectural simulation**

**Confidence:** 4
**Rating:** 7

**Detailed Feedback And Questions For Authors:**

While the paper makes claims on being able to predict performance metrics, a table that reports the same would be helpful. There is a brief qualitative discussion, please consider adding some quantitative metrics. Alternatively, it was not clear to me if the simulation errors only capture CPI error, or also errors in specific performance metrics.

**Top Reasons To Accept The Paper:**

This paper introduces TAO, a deep-learning based microarchitecture simulator designed to be faster and more efficient than existing approaches. TAO utilizes functional traces as input, allowing for reuse across different microarchitectures. It employs a multi-metric prediction model with self-attention to predict various performance metrics, including latency, branch mispredictions, and cache misses. To enable rapid exploration of different architectures, TAO incorporates transfer learning techniques using architecture-agnostic embedding layers. Evaluations show that TAO significantly reduces training and simulation time compared to state-of-the-art methods like SimNet and classical execution simulators like gem5.

Strengths:
* Uses functional traces as input and decouples microarchitecture from training of the DL model.
* Enables prediction of various performance metrics (like branch misprediction, cache miss), in addition to top-line metrics like time, CPI.
* Improved wall-clock runtime of simulation, without sacrificing fidelity.

**Top Reasons To Reject The Paper:**

While the paper makes claims on being able to predict performance metrics, a table that reports the same would be helpful. There is a brief qualitative discussion, please consider adding some quantitative metrics. Alternatively, it was not clear to me if the simulation errors only capture CPI error, or also errors in specific performance metrics.

---

### Official Review · Reviewer_NbD1 · 2024-05-30
**Review— TAO: Re-Thinking DL-based Microarchitecture Simulation**

**Confidence:** 4
**Rating:** 7

**Detailed Feedback And Questions For Authors:**

Overall, the paper was well-written and strong. There were many interesting and unique ideas for thinking about how representation learning should be done to elicit improved (ML-based) simulation results.

I had a couple questions and suggestions for potential improvement:

(1) Could you provide more details on your dataset in the paper? It is not clear to me as a reader at the moment exactly which benchmarks you generated SPEC traces from for training. It would be nice to know the size of the dataset to understand how much data is really needed to train these models.

(2) As mentioned, “As a common practice, architecture researchers often use popular software architecture simulators to incorporate their radical new ideas.” One of the benefits of a more traditional uArch simulator is the ability to modify the software and explore uArch design benefits. It is not clear to me right now from reading how an architect can easily do this in your current proposed set up. I understand there is the ability to easily perform transform learning on other uArchs but how would I evaluate a new uArch idea of mine? Would I need to first implement it in gem5, for example, collect a bunch of trace data, and then train a DL model? This seems like I would have already learned if my idea was good or not during the trace data collection but perhaps I am misunderstanding the work.

(3) PerfVec: “Learning Independent Program and Architecture Representations for Generalizable Performance Modeling” by Li et al. is another related work to consider.

(4) Looking at Figure 1, your evaluation compares (d) [your proposed TAO] to (a) [execution-driven simulator] and (c) [DL-based simulator] but not to (b) [trace-driven simulator]. Adding this to your evaluation would be helpful as a reader to put your results into context of the mechanisms you laid out.

**Top Reasons To Accept The Paper:**

This paper should be accepted: TOA proposes a new DL model for computer architecture simulation that shows significant throughput and training time improvements over SimNet, the other SOTA simulator. In addition to the model architecture, they highlight some fundamental principles to consider when building ML-based architecture simulators. This work is unique in its approach compared to the prior art in this space and is pushing on a very relevant topic that the community should be thinking about.

**Top Reasons To Reject The Paper:**

None.

---

### Decision · Program_Chairs · 2024-05-30

**Decision:**

Accept (Oral/Poster)

**Comment:**

Congratulations! We are pleased to inform you that your paper has been accepted for presentation at MLArchSys 2024. We look forward to your participation at the workshop. Further details regarding the schedule and format will be provided soon. See you at the workshop!